

# Competitive interactions between a nonmycorrhizal invasive plant, *Alliaria petiolata*, and a suite of mycorrhizal grassland, old field, and forest species

Gary T. Poon and Hafiz Maherali

Department of Integrative Biology, University of Guelph, Guelph, ON, Canada

## ABSTRACT

The widespread invasion of the nonmycorrhizal biennial plant, *Alliaria petiolata* in North America is hypothesized to be facilitated by the production of novel biochemical weapons that suppress the growth of mycorrhizal fungi. As a result, *A. petiolata* is expected to be a strong competitor against plant species that rely on mycorrhizal fungi for nutrient uptake services. If *A. petiolata* is also a strong competitor for soil resources, it should deplete nutrients to levels lower than can be tolerated by weaker competitors. Because the negative effect of losing the fungal symbiont for mycorrhizal plants is greatest when nutrients are low, the ability of *A. petiolata* to simultaneously suppress fungi and efficiently take up soil nutrients should further strengthen its competitive ability against mycorrhizal plants. To test this hypothesis, we grew 27 mycorrhizal tree, forb and grass species that are representative of invaded habitats in the absence or presence of competition with *A. petiolata* in soils that had previously been experimentally planted with the invader or left as a control. A history of *A. petiolata* in soil reduced plant available forms of nitrogen by >50% and phosphorus by 17% relative to control soil. Average mycorrhizal colonization of competitor species was reduced by >50% in *A. petiolata* history versus control soil. Contrary to expectations, competition between *A. petiolata* and other species was stronger in control than history soil. The invader suppressed the biomass of 70% of competitor species in control soil but only 26% of species in history soil. In addition, *A. petiolata* biomass was reduced by 56% in history versus control soil, whereas the average biomass of competitor species was reduced by 15%. Thus, our results suggest that the negative effect of nutrient depletion on *A. petiolata* was stronger than the negative effect of suppressing mycorrhizal colonization on competitor species. These findings indicate that the inhibitory potential of *A. petiolata* on competitor species via mycorrhizal suppression is not enhanced under nutrient limitation.

# INTRODUCTION

Invasions by exotic species are common and can negatively influence the structure and function of invaded communities and ecosystems (*Pimental et al., 2000*). Designing

Corresponding author
Hafiz Maherali,
maherali@uoguelph.ca

effective control and eradication programs to limit the spread of an invasive species, however, requires identifying the specific mechanism that facilitated invasion (*Mack et al., 2000*). Numerous mechanisms have been identified to explain successful invasions (*Catford, Jansson & Nilsson, 2009*; *Gurevitch et al., 2011*). For example, successful invaders may have high propagule production (*Colautti, Grigorovich & MacIsaac, 2006*), possess or evolve superior competitive ability for limiting resources (*Blossey & Notzold, 1995*), be released from specialist antagonists in their native range (*Callaway et al., 2004*), possess the ability to acclimate to a wide variety of conditions (*Parker, Rodriguez & Loik, 2003*), or secrete novel biochemical compounds that reduce the performance and survival of native inhabitants (*Callaway & Ridenour, 2004*).

Recent reviews suggest that successful invasions rarely occur because of a single mechanism (*Catford, Jansson & Nilsson, 2009*; *Gurevitch et al., 2011*). At least three explanations for weak effects of any one mechanism have been proposed (*Gurevitch et al., 2011*). First, the efficacy of a particular mechanism may depend on ecological context, where differences in resource availability and the functional attributes of resident species can either facilitate or increase resistance to invasion (*Funk et al., 2008*). Second, the importance of a particular mechanism could differ between phases of an invasion (*Dietz & Edwards, 2006*). For example, allelopathy may effectively suppress resident species in the initial phases of invasion (*Callaway & Ridenour, 2004*), but its effects can diminish as resident species acclimate or evolve resistance to the novel biochemicals (*Lankau et al., 2009*; *Lankau, 2011*). Third, multiple mechanisms could act synergistically, as observed in situations where invasion is facilitated by both competitive suppression of resident species and reduced palatability to herbivores (*Lau & Schultheis, 2015*). Simultaneous empirical assessments of multiple causes of invasion, however, are infrequent (*Zheng et al., 2015*).

The widespread invasion of *Alliaria petiolata* ((M. Bieb.) Cavara & Grandem, Brassicaceae), a biennial species native to Europe that was introduced to North America in the late 19th century (*Cavers, Heagy & Kokron, 1979*), has been attributed to several factors (*Rodgers, Stinson & Finzi, 2008*). Of these mechanisms, the ability of *A. petiolata* to produce allelopathic phytochemicals has received considerable attention. *A. petiolata* phytochemicals are present in leaf litter and also released as root exudates (*Cipollini et al., 2005*; *Rodgers, Stinson & Finzi, 2008*), but have limited direct negative effects on neighboring plant species (*McCarthy & Hanson, 1998*; *Roberts & Anderson, 2001*; *Prati & Bossdorf, 2004*; *Cipollini, Stevenson & Cipollini, 2008*). Instead, *A. petiolata* phytochemicals tend to suppress the growth of mycorrhizal fungi (*Roberts & Anderson, 2001*; *Stinson et al., 2006*; *Callaway et al., 2008*; *Rodgers, Stinson & Finzi, 2008*; *Wolfe et al., 2008*; *Cantor et al., 2011*), though the effect is variable (*Burke, 2008*; *Lankau et al., 2009*; *Lankau, 2011*). Because *A. petiolata* is non-mycorrhizal, whereas most plant species rely on mycorrhiza for nutrient uptake services (*Wang & Qiu, 2006*; *Brundrett, 2009*), the suppression of mycorrhizal fungi is expected to advantage *A. petiolata* relative to other species during establishment (*Stinson et al., 2006*; *Callaway et al., 2008*; *Hale & Kalisz, 2012*), though this effect can diminish over time (*Lankau et al., 2009*; *Lankau, 2011*).

The successful establishment and persistence of *A. petiolata* may also be influenced by the joint effects of novel biochemical weapons and the ability to acquire soil resources more effectively than potential competitors (*Blossey & Notzold, 1995*). Resource competition theory predicts that the strong competitors deplete limiting resources to levels lower than weaker competitors (*Tilman, 1988*; *Tilman & Wedin, 1991*; *Bever et al., 2010*). If *A. petiolata* is a strong resource competitor, it should deplete soil nutrients below tolerable limits for other species, suppressing other species more than itself. The ability to efficiently take up nutrients as well as survive and reproduce under limited soil nutrients could explain *A. petiolata*'s ability to colonize habitats that vary widely in nutrient availability (*Rodgers et al., 2008*) as well its ability to suppress native vegetation (e.g., *Stinson et al., 2007*; *Rodgers, Stinson & Finzi, 2008*). Moreover, if *A. petiolata* can suppress mycorrhizal fungi while simultaneously depleting soil nutrients, this should result in even stronger suppression of mycorrhizal competitors. This is because the negative effect of losing the fungal symbiont is greatest when mycorrhizal plants are grown in low soil nutrients (*Hoeksema et al., 2010*; *Johnson, 2010*).

Despite the potential for *A. petiolata* to modify the soil environment in a way that enhances its own competitive ability, resident species may still be able to resist invasion. Such resistance could depend on the morphological and physiological traits that influence acquisition of soil nutrients and light, which most often limit plant growth (*Grime, 1977*; *Gaudet & Keddy, 1988*; *Goldberg & Landa, 1991*; *Wardle et al., 1998*; *Funk et al., 2008*). The potential for depletion of soil nutrients and the suppression of mycorrhizal fungi by *A. petiolata* suggests that resident species which resist competitive suppression should have thin roots that maximize absorptive root surface area for resource uptake (*Goldberg, 1996*; *Casper & Jackson, 1997*). In addition, species that successfully resist invasion by *A. petiolata* could also be effective at acquiring light (*Stinson & Seidler, 2014*), particularly by accelerated height growth, which would allow them to overtop neighbors, and by having high photosynthetic light use efficiency (*Gaudet & Keddy, 1988*; *Goldberg & Landa, 1991*; *Rosch, Van Rooyen & Theron, 1997*; *Keddy et al., 2002*; *Wang et al., 2010*). The competitive ability of *A. petiolata* against other species has been tested in pairwise competition trials (*Meekins & McCarthy, 1999*; *Rodgers, Stinson & Finzi, 2008*; *Lankau, 2010*; *Leicht-Young, Pavlovic & Adams, 2012*), but whether growth in previously invaded soils enhances its competitive ability against mycorrhizal plant species is not known (*Hale & Kalisz, 2012*; *Smith & Reynolds, 2014*).

To test the hypothesis that nutrient depletion in the first year of an invasion enhances the competitive ability of *A. petiolata* against resident mycorrhizal plant species in subsequent years, we grew *A. petiolata* with and without multiple competitor species in forest soil that had either been left intact or previously planted with *A. petiolata*. This latter treatment simulates a reduction in soil nutrient availability because *A. petiolata* is expected to take up resources during growth. However, the experiment does not strictly mimic the entire process of invasion in the field because the high decomposability of *A. petiolata* leaves is expected to return nutrients to soil in the longer term (e.g., *Rodgers et al., 2008*). The experimental design nonetheless allows us to address whether the competitive ability of

*A. petiolata* can be influenced by changes in overall resource availability. Because *A. petiolata* occurs in a wide variety of habitats, including old fields, road sides, forest edges and forest understories (*Cavers, Heagy & Kokron, 1979*; *Stinson & Seidler, 2014*; *Smith & Reynolds, 2014*; *Biswas et al., 2015*), we quantified competition between *A. petiolata* and 27 native and non-native mycorrhizal competitor species that represent these different habitats (e.g., *Cavers, Heagy & Kokron, 1979*). We predicted that soil nutrient reduction and the potential for inhibition of mycorrhizal fungi by previous growth of *A. petiolata* in soil ('history soil') inhibits the growth of mycorrhizal plant species. As a result, *A. petiolata* should more strongly suppress the growth of, and resist growth suppression by, competitor species in the history soil treatment than in control soil. We predicted that competitor species with finer roots, greater height extension, and higher photosynthetic efficiency would be more likely to resist competition against *A. petiolata*.

## MATERIALS AND METHODS

To examine competitive interactions between resident species and *A. petiolata*, we grew 27 target species with and without the presence of *A. petiolata* (Table 1). Competitor species included forest trees, forest understory herbs, old field herbs and grasses that are commonly found in areas typically invaded by *A. petiolata* in southern Ontario (e.g., *Biswas et al., 2015*). *Alliaria petiolata* seeds were bulk collected from the Wild Goose Woods, a mixed hardwood forest in the University of Guelph Arboretum (43°32′N, 80°12′W) in July 2009. *Alliaria petiolata* can be found in dense patches along the periphery of the forest throughout this site. Seeds for each competitor species were harvested within the Guelph Arboretum as well as purchased from suppliers (Acorus Restoration, Walsingham, Ontario, Canada; Angelgrove Seed Company, Harbour Grace, Newfoundland and Labrador, Canada; Ontario Tree Seed Facility, Angus, Ontario, Canada; Richter's Herbs, Goodwood, Ontario, Canada (Table 1)).

To simulate a soil environment that *A. petiolata* is likely to encounter upon invasion, we grew plants in a forest soil without a history of *A. petiolata*. In November 2009, soil was collected to a depth of 30 cm from a mixed deciduous forest dominated by *Acer saccharum* in the Koffler Scientific Reserve (44°03′N, 79°29′W, Newmarket, ON). Prior to soil collection, live aboveground vegetation and macro-organic matter (leaves and twigs) were removed. Soil was sieved onsite to remove roots and stones and placed into 30, 35L tubs (60.7 cm long × 40.4 cm wide × 22.1 cm deep; Roughneck Storage Box #2214; Newell Rubbermaid Inc., Atlanta, Georgia, USA). Tubs had holes drilled in the bottom to facilitate drainage. Soils were stored at 4 °C prior to the beginning of experiments. To experimentally create a treatment where the presence of *A. petiolata* has modified the soil, we grew *A. petiolata* plants in half of the field collected soil (e.g., *Callaway et al., 2008*). The remaining soil was left intact in the tubs. To create the *A. petiolata* soil history treatment, *A. petiolata* seeds were cold stratified at 4 °C for 120 days on moist filter paper placed inside 10 cm diameter parafilm sealed petri dishes. In January 2010, 100 germinating *A. petiolata* seeds were transplanted into each of 15 randomly selected tubs. After 6 weeks, seedlings were thinned to a density of 80 plants/m$^2$, which approximates the upper end

**Table 1 List of competitor species used in the study, along with information on their plant family affiliation, growth form, status in North America (18 native, 9 introduced), and whether plants are arbuscular mycorrhizal (AM), ecto-mycorrhizal (ECM), or ambiguous (both mycorrhizal and non-mycorrhizal states reported in the literature).** Mycorrhizal state was determined from *Wang & Qiu (2006)*.

| Latin name | Family | Growth form | Status | Mycorrhizal state |
|---|---|---|---|---|
| *Acer saccharum* L. | Aceraceae | Tree | Native[c] | AM |
| *Juglans nigra* L. | Juglandaceae | Tree | Native[c] | AM |
| *Pinus strobus* L. | Pinaceae | Tree | Native[c] | ECM |
| *Prunus virginiana* L. | Rosaceae | Tree | Native[c] | AM |
| *Quercus macrocarpa* Michx. | Fagaceae | Tree | Native[c] | ECM |
| *Thuja occidentalis* L. | Cupressaceae | Tree | Native[c] | AM |
| *Achillea millefolium* L. | Asteraceae | Perennial Forb | Native[d] | AM |
| *Aquilegia vulgaris* L. | Ranunculaceae | Perennial Forb | Introduced[d] | AM |
| *Aster umbellatus* Miller | Asteraceae | Perennial Forb | Native[b] | AM |
| *Daucus carota* L. | Apiaceae | Biennial Forb | Introduced[e] | AM |
| *Fragaria virginiana* Miller. | Rosaceae | Perennial Forb | Native[b] | AM |
| *Hesperis matronalis* L. | Brassicaceae | Biennial Forb | Introduced[a] | Ambiguous |
| *Hypericum perforatum* L. | Clusiaceae | Perennial Forb | Introduced[a] | AM |
| *Leucanthemum vulgare* Lam. | Asteraceae | Perennial Forb | Introduced[a] | AM |
| *Lobelia siphilitica* L. | Campanulaceae | Perennial Forb | Native[a] | AM |
| *Plantago lanceolata* L. | Plantaginaceae | Perennial Forb | Introduced[d] | AM |
| *Prunella vulgaris* L. | Lamiaceae | Perennial Forb | Native[b] | AM |
| *Rudbeckia hirta* L. | Asteraceae | Perennial Forb | Native[b] | AM |
| *Sambucus nigra spp. canadensis* L. | Caprifoliaceae | Perennial Forb | Native[a] | Ambiguous |
| *Solidago canadensis* L. | Asteraceae | Perennial Forb | Native[e] | AM |
| *Taraxacum officinale* F.H. Wigg. | Asteraceae | Perennial Forb | Introduced[b] | AM |
| *Trifolium pratense* L. | Fabaceae | Biennial Forb | Introduced[e] | AM |
| *Bromus inermis* Leyss. | Poaceae | Perennial Grass | Introduced[e] | AM |
| *Elymus canadensis* L. | Poaceae | Perennial Grass | Native[b] | AM |
| *Elymus riparius* Wiegand. | Poaceae | Perennial Grass | Native[b] | AM |
| *Elymus virginicus* L. | Poaceae | Perennial Grass | Native[b] | AM |
| *Panicum virgatum* L. | Poaceae | Perennial Grass | Native[a] | AM |

**Notes.**

[a] Seeds obtained from Acorus Restoration.
[b] Seeds obtained from Angelgrove Seed Company.
[c] Seeds obtained from Ontario Tree Seed Facility.
[d] Seeds obtained from Richters Herbs, or field collections.
[e] Seeds obtained from University of Guelph Arboretum.

of *A. petiolata* density in field populations (*Meekins & McCarthy, 2002*). Tubs containing *A. petiolata* seedlings and those containing intact control soil were randomly arranged on the greenhouse bench and watered to maintain field capacity. Germinating seedlings from other species were periodically removed from all tubs. *Alliaria petiolata* plants were harvested 5 months after germination to simulate the approximate active growing season for first year rosettes of this species in southern Ontario and to allow roots to fully explore the soil in the tubs. At harvest, the aboveground portion of plants was removed and discarded, and soil was sieved to remove roots and homogenized within each soil treatment.

To quantify the effect of *A. petiolata* history on soil nutrients, we sampled 500 mL of soil from the post-harvest homogenized soil mixture for each treatment and analyzed it for plant available nutrients, including $NO_3^-$, $NH_4^+$, P (Olsen), Mg, and K in mg per unit mass (kg) or volume (L) of soil (University of Guelph Laboratory Services; www.guelphlabservices.com/AFL/plants.aspx). Soil that had been left without *A. petiolata* contained 160 mg/kg $NO_3^-$, 18.3 mg/kg $NH_4^+$, 23 mg/L P, 77 mg/L Mg, and 52 mg/L K. In soil with *A. petiolata* history, these amounts were reduced to 29.2 mg/kg for $NO_3^-$ (−82% decrease), 8.56 mg/kg for $NH_4^+$ (−53% decrease), 19 mg/L for P (−17% decrease), 53 mg/L for Mg (−31% decrease) and 40 mg/L for K (−23% decrease).

To study the effects of soil treatment, interspecific competition and competitive species identity on the growth of either *A. petiolata* or the competitor, we used a three-factor design. To quantify competition, we grew each competitor species in the presence and absence of an *A. petiolata* individual in the same pot (e.g., *Gaudet & Keddy, 1988*; *Wang et al., 2010*) in both soil treatments. We also grew *A. petiolata* alone as a reference to calculate its response to competition with the other species. Each treatment combination ((27 species + *A. petiolata*) × 2 soil treatments × 2 competition treatments) was replicated 6 times for a total of 744 pots. Pots (650 mL volume, 6.4 cm wide × 25 cm deep; D40 R; Stuewe and Sons Inc., Tangent, Oregon, USA) were filled with either *A. petiolata* history or untreated field soil and were randomly arranged in a checkerboard pattern across 53 trays (57 N25T; Stuewe and Sons Inc., Tangent, Oregon, USA) to minimize competition for light between pots. To induce germination, all seeds were cold stratified for 30–120 days based on information provided by seed suppliers. Cold stratification times were staggered to ensure all species germinated at the same time. After stratification, seeds were moved to the University of Guelph Phytotron greenhouse and germinated in a medium of 2/3 top soil and 1/3 silica sand and then transplanted into experimental treatments. Because of slow germination in some species, they were planted in two groups separated by two weeks. All plants were grown for the same number of days and were completely randomized across the greenhouse benches. 100 mL of 1/4 strength 18-9-18 N:P:K fertilizer (Plant Products, Leamington, Ontario, Canada) was added once to all pots in both soil treatments to promote seedling establishment. Because an equal amount of fertilizer was added to all pots, nutrient differences between the *A. petiolata* history and control soil treatments remained. After 63 days, when herbaceous competitor plants had reached reproductive maturity, the aboveground parts of plants were harvested, separated according to species, dried at 60 °C for 48 h and weighed.

To determine if *A. petiolata* soil history suppressed arbuscular mycorrhizal (AM) fungi, we harvested the roots of a subset of competitor species when grown alone in both history and control soil. To quantify root colonization by AM fungi, we selected eight species that represented the range of growth forms in the experiment. Root cell contents were cleared with potassium hydroxide and AM fungi were stained with Chlorazol black E (*Brundrett, Piche & Peterson, 1984*). Samples were mounted on glass slides and viewed under a compound microscope at 250× magnification. To quantify fungal colonization by AM hyphae, arbuscules, and vesicles, we used the gridline intersection

method (*McGonigle et al., 1990*). Colonization was quantified as the presence or absence of well-stained structures at 50 intersections per root sample.

To determine whether morphological and physiological traits could explain the ability of competitive species to either resist suppression by, or suppress *A. petiolata*, we measured aboveground traits on all plant species in the absence of competition in both soil treatments. We measured leaf chlorophyll concentration and photosynthetic efficiency of up to 6 individuals from each species in each soil treatment at 5 and 9 weeks growth. Height at 5 weeks on these individuals was recorded as the vertical distance from the soil surface to the tip of the tallest leaf. We measured chlorophyll concentration on the three youngest fully expanded leaves per plant using a portable chlorophyll meter (SPAD 502; Minolta Inc., Ramsey, New Jersey, USA), and calculated an average value per plant. We measured photosynthetic efficiency as instantaneous fluorescence yield under saturating light conditions ($1,500 \, \mu mol \, m^{-2} \, s^{-1}$), a measure of the light use efficiency of photosystem II (*Maxwell & Johnson, 2000*). The three youngest fully expanded leaves per plant were measured using a light-adapted fluorometer (PAM-2500; Heinz Walz APbH, Effeltrich, Germany) and an average value per plant was calculated.

To determine if root traits co-varied with competitive ability, we grew 5 replicates of all plant species in a separate experiment in a sterilized mixture of 2/3 silica sand and 1/3 topsoil for 35 days. Root architecture could not be measured in the main experiment because roots could not be effectively separated from the higher organic matter containing field soil, and because roots were becoming pot bound by the time of harvest. The shorter growing period and silica sand-topsoil mixture prevented plant roots from becoming pot bound and facilitated the harvest of intact root systems. Plants were grown individually in 650 mL pots (D40 R; Stuewe and Sons Inc., Tangent, Oregon, USA). At harvest, roots were cleaned and preserved in 50% ethanol. For analysis, roots were stained with 0.05% Toluidine Blue O to improve the visibility of fine roots, spread out in water to minimize overlap and photographed with a high resolution (6,400 dpi) scanner (Epson V700; Epson Canada Limited, Markham, Ontario, Canada). Root images were analyzed with WinRhizo software (version 2009a; Regent Instruments 2009, Quebec City, Quebec, Canada) using the automatic pixel classification setting to assess the length and average root diameter of each root system. After scanning, roots were dried at 60 °C for 48 h and weighed. In addition to average root diameter, we also calculated specific root length (SRL), or the ratio of root length to root mass, which is indicative of the amount of surface area available for nutrient absorption (*Craine et al., 2001*).

To assess the magnitude and variation in resistance of competitor species to *A. petiolata* competition and whether the magnitude of resistance is influenced by *A. petiolata* soil history, we analyzed aboveground biomass of competitor species with a three-way analysis of variance (ANOVA) with competition, soil history and competitor species identity and all interactions as factors. Planned orthogonal single degree of freedom (1-df) contrasts were used to determine whether each competitor species biomass differed between competition treatments within each soil history treatment. We also used 1-df contrasts to test whether

growth forms (trees, forbs, grasses) differed as a whole between competition treatments in each soil history treatment.

To assess the magnitude and variation in the ability of competitor species to influence *A. petiolata* aboveground biomass, and whether this species effect was influenced by soil history, we used a two-way ANOVA with competitor species identity, soil history and their interaction as factors. To test whether growth with a competitor species suppressed the biomass of *A. petiolata* in each soil treatment, we used planned orthogonal 1-df contrasts to compare the biomass of *A. petiolata* grown alone relative to its growth (i) with each competitor, (ii) with each growth form in aggregate, and (iii) across all competitor species in aggregate. The effect of soil treatment on fungal colonization of roots was determined with a two-way ANOVA with soil and species as factors. The statistical significance of soil treatment on fungal colonization for each species was determined by comparing 95% confidence intervals for overlap. The effect of growth form and soil treatment on plant traits was tested with a two-way ANOVA using species means for each trait as the replicate. Differences among growth forms were determined by comparing the 95% confidence intervals for each growth form for overlap following a significant main effect. All ANOVAs and 1-df contrasts were done with SPSS 22.0 (IBM Corp., Armonk, NY).

To quantify variation in the ability of competitor species to either resist suppression by *A. petiolata* or suppress *A. petiolata*, we calculated two indices of competition. The ability of a competitor species to resist suppression is defined as competitive response (CR, *Wang et al., 2010*), and was quantified as ln (biomass under competition/biomass alone). The ability of each competitor species to suppress *A. petiolata* is defined as competitive effect (CE, *Gaudet & Keddy, 1988*; *Wang et al., 2010*), and was quantified as –ln (*A. petiolata* biomass under competition/*A. petiolata* biomass alone). When calculated this way, greater values reflect stronger competitive ability.

To determine whether morphological and physiological traits of competitor plants were associated with competitive ability, we used phylogenetic generalized least squares (PGLS) multiple regression, with competitive ability (either CR or CE) as the dependent variable and traits as independent variables. Growth form of plants was used as a covariate in the analysis. Because root traits were assessed in a different experiment, multiple regression analyses were run separately for aboveground and belowground traits. To analyze data, we used the time calibrated phylogenetic tree from *Davies et al. (2004)* in Phylomatic (*Webb, Ackerly & Kembel, 2008*), pruned to include the competitor species. In PGLS regression, the phylogenetic variance–covariance matrix is incorporated into the calculation of coefficients ($\beta$) for either a univariate or multiple regression model (*Martins & Hansen, 1997*; *Pagel, 1999*). To calculate the magnitude of phylogenetic effects on the regression, maximum likelihood is used to estimate $\lambda$, an index which varies from 0, indicating complete independence between variation in the regression residuals and phylogeny, and 1, indicating complete dependence between residual variation with Brownian model of evolution (*Freckleton, Harvey & Pagel, 2002*). When $\lambda = 0$, the PGLS regression is identical to ordinary least squares regression. PGLS regression and estimates of $\lambda$ were done in R

**Table 2** A three-way ANOVA table describing the effects of species identity, competition with *A. petiolata*, soil history and their interactions on dry mass of competitor species.

| Source | Type III sums of squares | df | Mean square | F | P |
|---|---|---|---|---|---|
| Species | 1141.51 | 26 | 43.90 | 66.78 | $5.29 \times 10^{-144}$ |
| Soil history | 1.41 | 1 | 1.41 | 2.14 | 0.144 |
| Competition | 209.97 | 1 | 209.97 | 319.38 | $1.57 \times 10^{-55}$ |
| Species * Soil history | 32.20 | 26 | 1.24 | 1.88 | 0.006 |
| Species * Competition | 141.18 | 26 | 5.43 | 8.26 | $1.76 \times 10^{-25}$ |
| Soil history * Competition | 45.14 | 1 | 45.14 | 68.65 | $1.10 \times 10^{-15}$ |
| Species * Soil history * Competition | 68.44 | 26 | 2.63 | 4.00 | $4.42 \times 10^{-10}$ |
| Error | 326.75 | 497 | 0.66 | | |

version 3.12 (*R Core Team, 2014*) using the 'pgls' command in the package caper, version 0.5.2 (*Orme et al., 2013*).

## RESULTS

The average aboveground biomass of competitor species was reduced by the presence of *A. petiolata* compared to when they were grown alone (significant competition main effect, Table 2). However, competition was weaker in soil with a history of *A. petiolata* relative to control soil (significant soil history × competition interaction, Table 2 and Fig. 1). The average biomass of competitor species was reduced by 59% in control soil compared to 27% in *A. petiolata* history soil. The influence of soil history on competition also varied among species (significant species × soil history × competition interaction, Table 2 and Fig. 1). For example, the aboveground biomass of 19 species (70%; 13/18 native, 6/9 introduced) was suppressed by competition in control soil whereas only 7 species (26%; 4/18 native, 3/9 introduced) were suppressed by competition in *A. petiolata* history soil. On a growth form basis, trees ($-38\%$, $P = 0.01$), forbs ($-62\%$, $P < 0.000001$) and grasses ($-56\%$, $P < 0.000001$) were all suppressed by *A. petiolata* in control soil, whereas the biomass of forbs ($-32\%$, $P < 0.000001$) and grasses ($-14\%$, $P = 0.007$), but not trees ($P = 0.228$), was significantly reduced by competition in the *A. petiolata* history soil (Fig. 1, insets).

The average biomass of *A. petiolata* in competition was not significantly different from its average biomass when grown alone in either soil treatment ($P_{history} = 0.284$, $P_{control} = 0.602$; Fig. 2). *Alliaria petiolata* biomass varied in response to competition with different competitor species (significant species effect, Table 3 and Fig. 2). In most cases, these species effects were not consistent between control and history soils (significant species × soil history interaction, Table 3). For example, relative to its biomass when alone, *A. petiolata* was significantly smaller in competition with *H. matronalis, B. inermis, E. canadensis, E. riparius*, and *E. virginicus* in control soil but significantly smaller in competition with *Q. macrocarpa, H. matronalis* and *E. canadensis* in history soil (Fig. 2). In some cases, *A. petiolata* biomass was higher when grown with a competitor species than when grown alone. This response occurred with *P. strobus* and *T. occidentalis* in control soil and *H. perforatum* in history soil. *A. petiolata* biomass response to competition also

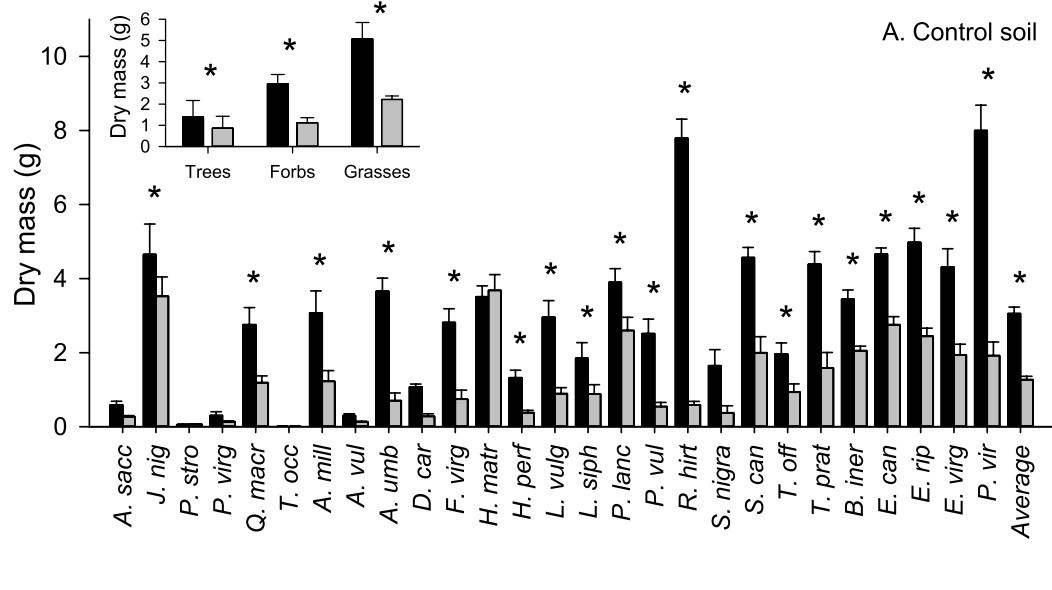

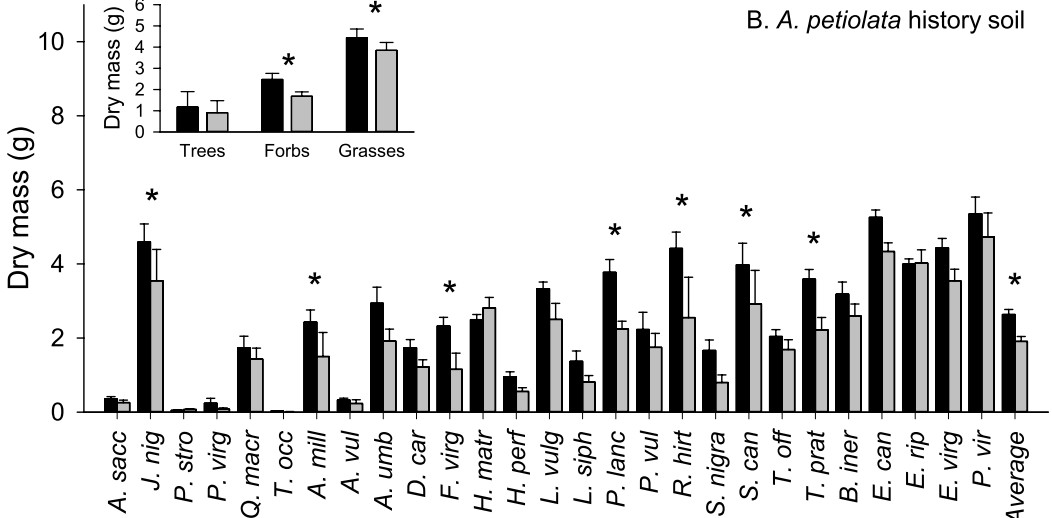

**Figure 1** **Biomass of competitor species in response to competition with *A. petiolata* in control (A) or soil with a history of *A. petiolata* (B).** Biomass within each growth form are shown in the insets. Black bars indicate plants grown alone and grey bars indicate plants grown in competition with *A. petiolata*. Statistically significant differences were determined using planned orthogonal 1-df contrasts, and are indicated with an asterisk.

varied with growth form, and this effect differed between soil treatments (Fig. 2, insets). In control soil, *A. petiolata* biomass was 27% higher ($P = 0.041$) when grown with trees than when grown alone, 43% lower ($P = 0.001$) when grown with grasses than when grown alone, and not influenced by forbs ($P = 0.61$). In history soil, *A. petiolata* biomass was not affected by competition with trees ($P = 0.96$) or forbs ($P = 0.38$), but was 62% lower ($P = 0.033$) when grown with grasses than when grown alone.

On average, competitor species grown alone in soil with a history of *A. petiolata* were 15% smaller than plants grown in control soil ($F_{1,261} = 21.991$, $P = 0.000004$, Fig. 3).

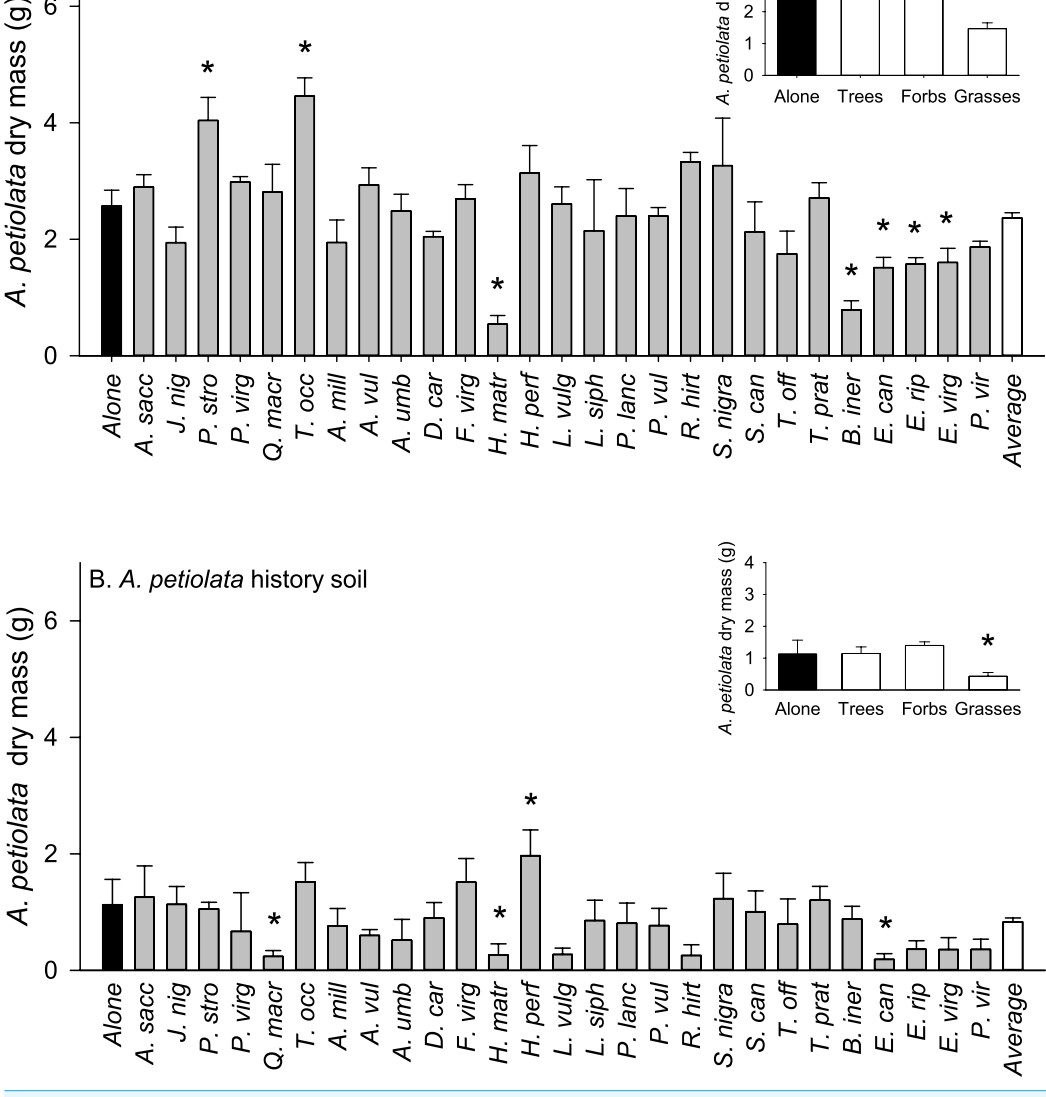

**Figure 2** **Biomass of *A. petiolata* alone or in response to competition with other species in control (A) or soil with a history of *A. petiolata* (B).** Biomass of *A. petiolata* alone versus in competition with members of different growth forms are shown in the insets. Statistically significant differences were determined using planned orthogonal 1-df contrasts, and are indicated with an asterisk.

Competitor species also differed in their response to *A. petiolata* soil history ($F_{26,261} = 3.042$, $P = 0.000003$), though a majority showed no significant difference between treatments. Significant negative effects of soil history were found for *Q. macrocarpa* ($-37\%$, $P = 0.041$), *H. matronalis* ($-29\%$, $P = 0.041$), *R. hirta* ($-43\%$, $P < 0.000001$), *E. riparius* ($-20\%$, $P = 0.048$), and *P. virgatum* ($-33\%$, $P < 0.000001$). The strongest negative response to soil history was observed for *A. petiolata*, whose biomass was 56% lower in soil in which it had been previously planted than in control soil ($P = 0.004$).

**Table 3  A two-way ANOVA table describing the effects of competitor species identity, soil history and their interaction on the dry mass of *A. petiolata*.**

| Source | Type III sums of squares | df | Mean square | F | P |
|---|---|---|---|---|---|
| Species | 92.4 | 26 | 3.55 | 6.76 | $3.74 \times 10^{-17}$ |
| Soil history | 179.59 | 1 | 179.59 | 341.39 | $9.51 \times 10^{-48}$ |
| Species * Soil history | 40.37 | 26 | 1.55 | 2.95 | $7.30 \times 10^{-06}$ |
| Error | 124.15 | 236 | 0.53 | | |

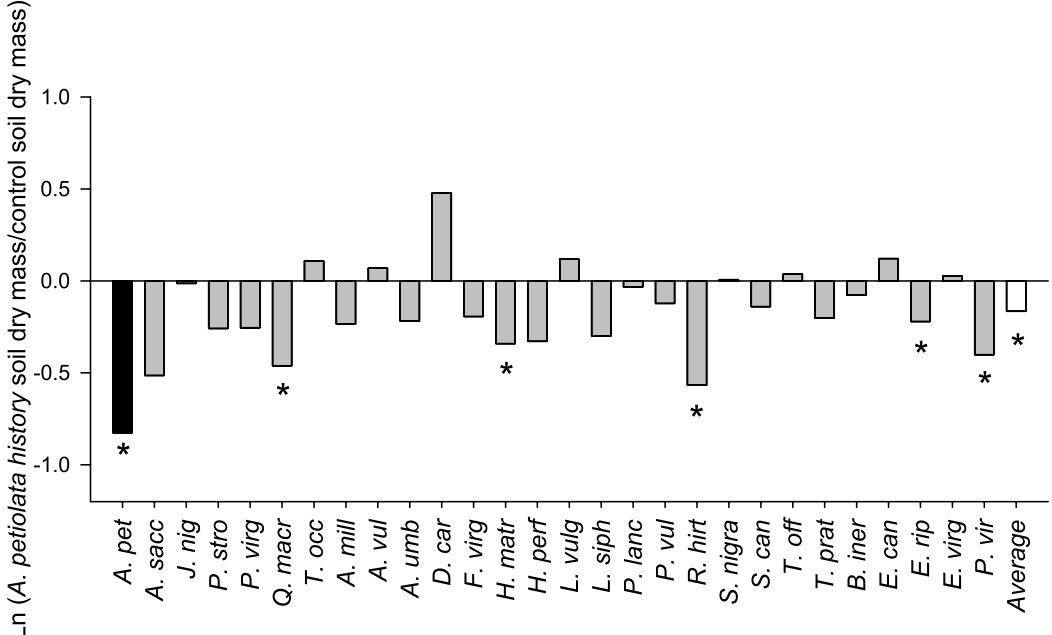

**Figure 3  The log response ratio of plant biomass without competition in *A. petiolata* history relative to control soil.** Statistically significant differences between soil treatments were determined using planned orthogonal 1-df contrasts, and are indicated with an asterisk.

Plants grown in soils with a history of *A. petiolata* had reduced levels of arbuscular mycorrhizal colonization of roots (Fig. 4). On average, plants in the soil history treatment had 57% reduced hyphal colonization ($F_{1,60} = 20.47$, $P = 0.000029$), 53% reduced arbuscular colonization ($F_{1,60} = 4.97$, $P = 0.029$), and 57% reduced vesicular colonization ($F_{1,60} = 4.95$, $P = 0.030$) than plants grown in control soils. These effects were strongest in *Q. macrocarpa*, *F. virginiana* and *E. canadensis* for hyphae (Fig. 4A), *H. perforatum* for arbuscles (Fig. 4B) and *F. virginiana* for vesicles (Fig. 4C). We note that *Q. macrocarpa* is not typically colonized by AM fungi (Table 1), and so the levels of fungal colonization reported for this species may reflect a non-functional symbiosis. The average sizes of the soil history effect on colonization with *Q. macrocarpa* removed from the dataset were −53% for hyphae ($F_{1,51} = 14.7$, $P = 0.000353$), −52.5% for arbuscles ($F_{1,51} = 4.63$, $P = 0.036$) and −54.5% for vesicles ($F_{1,51} = 3.38$, $P = 0.072$).

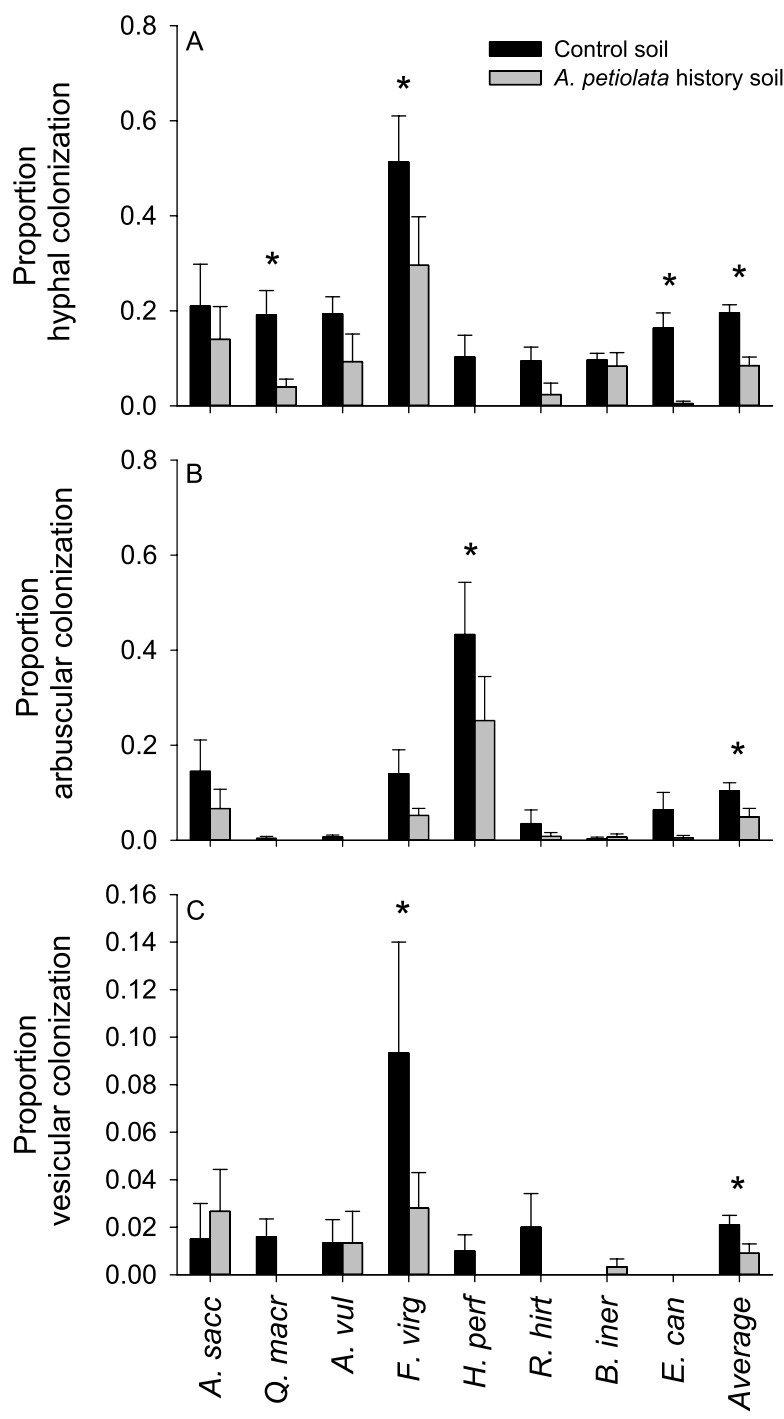

**Figure 4 The effect of soil history with _A. petiolata_ on the colonization of roots by arbuscular mycorrhizal (AM) hyphae (A) AM arbuscules (B), and vesicles (C).** Statistically significant differences between soil treatments are indicated with an asterisk.

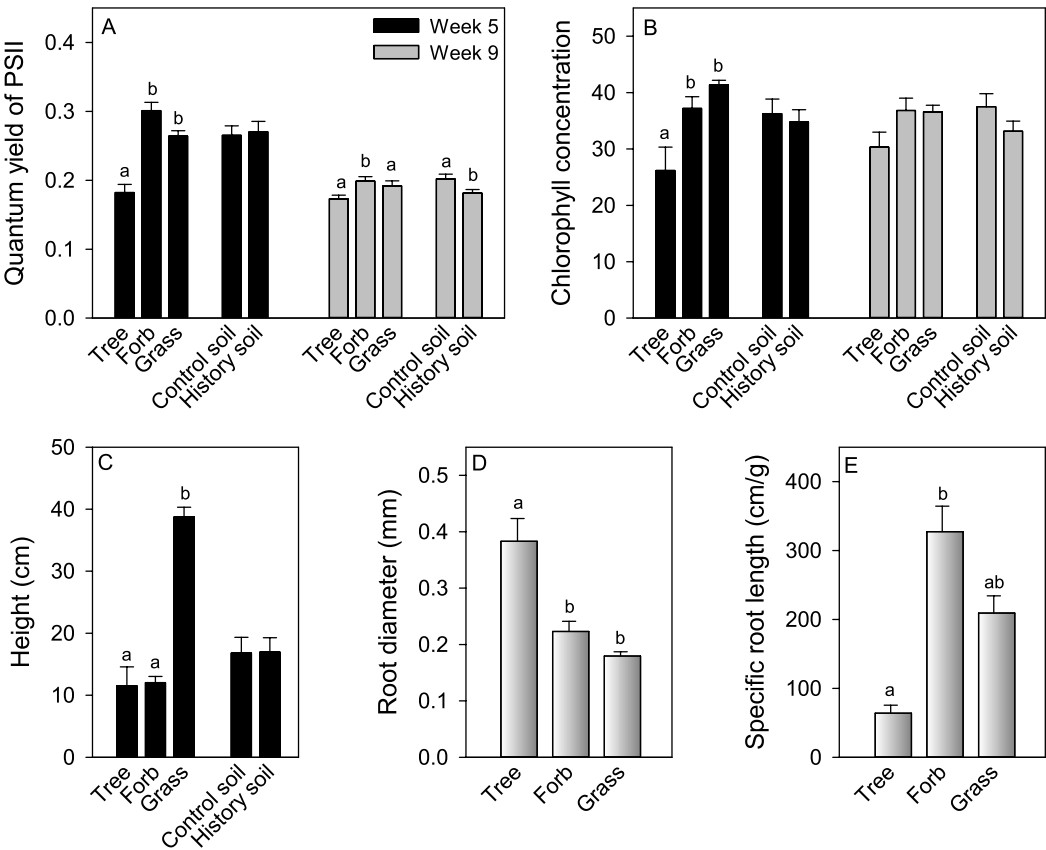

**Figure 5** **The effect of growth form and exposure to either control or *A. petiolata* soil history on quantum yield of PSII at weeks 5 and 9 (A), leaf chlorophyll concentration at weeks 5 and 9 (B) and plant height at week 5 (C).** The effect of growth form on root diameter (D) and specific root length (E). Different letters above bars, when present, represent statistically significant differences ($P < 0.05$) among groups within each treatment, as determined by a comparison of 95% confidence limits among groups.

Morphological and physiological traits of competitor plants grown alone differed among growth forms, but were not generally affected by growing in soil with a history of *A. petiolata* (Fig. 5). Quantum yield of photosystem II [Y(II)] measured at week 5 was significantly higher in herbs and grasses relative to trees. In week 9, Y(II) was significantly higher in herbs compared to grasses and trees (Fig. 5A). The quantum yield of photosystem II did not differ between soil treatments in week 5, but was lower in *A. petiolata* history soil than control soil in week 9. Chlorophyll concentration in week 5 was significantly higher in herbs and grasses than trees, but did not differ among growth forms in week 9, and did not differ between soil history treatments (Fig. 5B). At week 5, grasses were significantly taller than trees and herbs, but height was not influenced by soil history treatment (Fig. 5C). Trees had significantly larger root diameter than either herbs or grasses, which did not differ significantly from each other. Trees also had significantly lower SRL than herbs, whereas grasses had intermediate SRL that did not differ significantly from that of trees or herbs (Figs. 5D and 5E).

**Table 4 Partial correlation coefficients (β) indicating relationships between competitive response (CR) or competitive effect (CE) in control or history soil, and plant functional traits, including height at 5 weeks, quantum yield of PS II in the light [Y(II)] at 5 and 9 weeks, leaf chlorophyll content at 5 and 9 weeks, mean root diameter and specific root length (SRL).** Because traits differed between trees, forbs and grasses, plant growth form was included as a covariate in the analysis, but only β and significance values for traits are shown. The degree to which residuals from the multiple regression were correlated with phylogeny is indicated by λ.

| Dependent variable | Trait | β | P | Dependent variable | Trait | β | P |
|---|---|---|---|---|---|---|---|
| CR control soil | Height @ 5 wks | −0.12 | 0.59 | CR history soil | Height @ 5 wks | −0.09 | 0.70 |
| λ = 0 | Y(II) @ 5 wks | 0.023 | 0.92 | λ = 1 | Y(II) @ 5 wks | 0.42 | 0.06 |
| | Y(II) @ 9 wks | 0.18 | 0.44 | | Y(II) @ 9 wks | 0.082 | 0.72 |
| | Chl @ 5 wks | −0.040 | 0.86 | | Chl @ 5 wks | 0.080 | 0.73 |
| | Chl @ 9 wks | 0.053 | 0.82 | | Chl @ 9 wks | −0.16 | 0.49 |
| CE control soil | Height @ 5 wks | −0.02 | 0.93 | CE history soil | Height @ 5 wks | 0.064 | 0.78 |
| λ = 0 | Y(II) @ 5 wks | 0.27 | 0.24 | λ = 0 | Y(II) @ 5 wks | 0.063 | 0.79 |
| | Y(II) @ 9 wks | −0.11 | 0.64 | | Y(II) @ 9 wks | 0.050 | 0.83 |
| | Chl @ 5 wks | 0.061 | 0.79 | | Chl @ 5 wks | −0.12 | 0.61 |
| | Chl @ 9 wks | 0.11 | 0.63 | | Chl @ 9 wks | 0.25 | 0.27 |
| CR control soil | Root diameter | −0.23 | 0.28 | CR history soil | Root diameter | −0.11 | 0.61 |
| λ = 0 | SRL | 0.09 | 0.67 | λ = 0.981 | SRL | −0.13 | 0.57 |
| CE control soil | Root diameter | −0.35 | 0.11 | CE history soil | Root diameter | −0.22 | 0.32 |
| λ = 1 | SRL | −0.10 | 0.64 | λ = 0 | SRL | −0.23 | 0.30 |

Though above and belowground functional traits varied among growth forms, these traits were generally not associated with their ability to compete in either soil environment, measured as either the ability to resist suppression from (competitive response, CR) or suppress (competitive effect, CE) *A. petiolata* (Table 4). The only exception to this pattern was the nearly significant ($P = 0.06$) positive relationship between Y(II) @ 5 weeks and competitive response in *A. petiolata* history soil. In addition, even though species varied in their response to soil history, this variation was also not correlated with competitive ability. The ln response ratio of growth in *A. petiolata* history versus control soils (Fig. 3) was not associated with either competitive response ($F_{1,25} = 0.07$, $r^2 = 0.003$, $P = 0.79$, $\lambda = 0.978$) or competitive effect ($F_{1,25} = 0.57$, $r^2 = 0.022$, $P = 0.45$, $\lambda = 0.266$).

Metrics of competitive ability were not strongly correlated across soil treatments. Specifically, CR in control soil only explained 5.7% of the variation in CR in *A. petiolata* history soil, and CE in control soil only explained 12% of the variation in CE in history soil (Fig. 6). CR and CE were positively correlated in control soil ($F_{1,25} = 7.44$, $r^2 = 0.229$, $P = 0.01$, $\lambda = 1$), but were not correlated with each other in history soil ($F_{1,25} = 2.73$, $r^2 = 0.098$, $P = 0.11$, $\lambda = 0$).

## DISCUSSION

Our results indicate that *A. petiolata* is a strong competitor against a range of common mycorrhizal grassland, old field and forest species in soils which had no previous history with its conspecifics, but contrary to expectation, this competitive advantage weakens in soil with a history of conspecific growth. In uninvaded control soil, for example, *A. petiolata* suppressed the biomass of a majority of competitor plant species by an

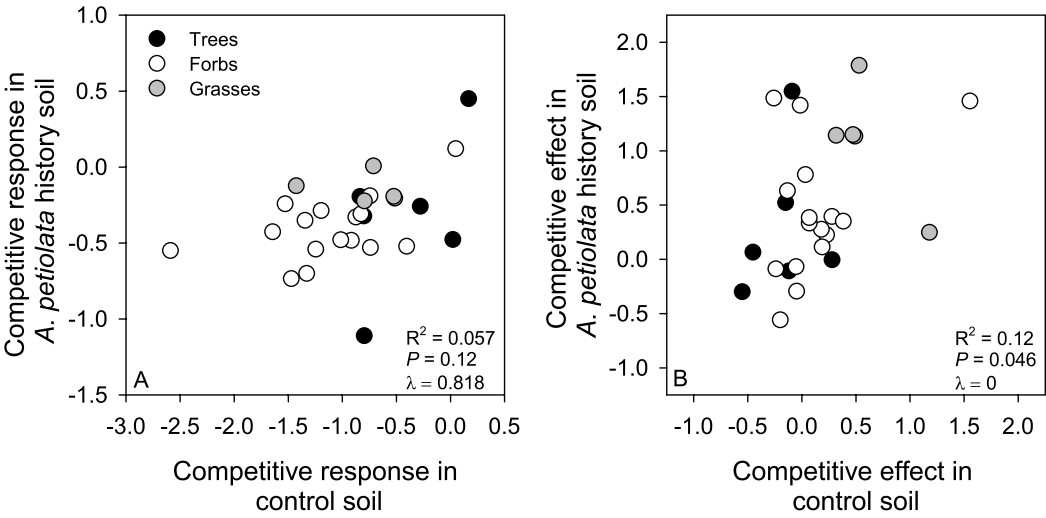

**Figure 6** Relationships between competitive response (A) or competitive effect (B) across control and *A. petiolata* history soils.

average effect size that exceeded 50% (Fig. 1). By contrast, the suppression of competitor species' biomass by *A. petiolata* was weaker in history soil, with an effect size that was less than half of that observed in control soil. Moreover, 70% of species responded negatively to the presence of *A. petiolata* in control soil, but only 26% of species responded negatively in conspecific history soil. Because the soil history treatment reduced plant available nutrients in soil and reduced mycorrhizal colonization of roots, differences in *A. petiolata* competitive ability between treatments could be caused by these factors acting independently or in combination. Nevertheless, our findings suggest that the competitive ability of newly introduced *A. petiolata* is sufficient to displace competitor species in previously uninvaded sites in the short term, but modification of the soil environment by *A. petiolata* may not enhance its competitive ability.

The weaker competitive effect of *A. petiolata* on other species in history soil occurred despite suppression of mycorrhizal fungi. Mycorrhizal colonization in soils with *A. petiolata* history (Fig. 4) was reduced by levels comparable to that observed in the field (e.g., *Barto et al., 2011*), with concomitant reductions in competitor plant growth (Fig. 3). However, competitor species were still better able to resist competition from *A. petiolata* in soils with a history of the invader than control soils. Though we cannot separate the individual effects of nutrient depletion and reduced mycorrhizal colonization on the outcome of competition in the present study, we note that *A. petiolata* was suppressed in the soil history treatment at a level that was more than three times the average level of suppression across all competitor species (Fig. 3). Because *A. petiolata* is nonmycorrhizal, the strong negative effect of growth in conspecific soil on its biomass was most likely caused by lower nutrients. We suggest, therefore, that the most likely explanation for weaker competitive ability of *A. petiolata* in history soils is that the negative effect of nutrient depletion on *A. petiolata* was stronger than the negative effect of suppressing mycorrhizal colonization on competitor species. *Davis et al. (2012)* also observed weak effects of *A. petiolata* soil

history on the biomass of competitor species. The observation that competition was weaker overall in history soil is also consistent with the hypothesis that when plant growth is suppressed by environmental stress or low fertility, limited overall demand for resource uptake reduces the strength of competition (*Grime, 1977*; *Lamb, Shore & Cahill, 2007*).

Our findings imply that the negative effect of *A. petiolata* on mycorrhizal fungi as a mechanism of competition during invasion may be weaker than previously expected. Though this interpretation is supported by weaker competition in the *A. petiolata* history soil, where colonization of roots by AM fungi was reduced, it is tentative for two reasons. First, we did not quantify AM fungal colonization of roots for all competitor species, and it is possible that these effects were not the same in unmeasured species. Second, though the reduction in AM fungal colonization of roots is consistent with the presence of fungal inhibiting secondary chemicals produced by *A. petiolata*, we did not directly quantify the concentration of these compounds in *A. petiolata* history soil. Though secondary chemicals produced by *A. petiolata* likely reduced fungal populations during the 5 month soil conditioning period (*Roberts & Anderson, 2001*; *Stinson et al., 2006*), they may have been absent in the main experiment because of a short half-life (*Barto & Cipollini, 2009*).

We hypothesized that *A. petiolata*'s strong competitive ability is caused by the capacity to deplete limiting resources to levels lower than resident species, as expected from resource competition theory (*Tilman, 1988*; *Tilman & Wedin, 1991*; *Bever et al., 2010*). However, simulated nutrient depletion of soils by *A. petiolata* more strongly suppressed its own growth relative to competitor species. The ability of *A. petiolata* to suppress other species may therefore be caused by other traits, such as fast growth rate and high allocation to leaf area (*Grime, 1977*; *Funk et al., 2008*; *Engelhardt & Anderson, 2011*). The tendency for decomposing *A. petiolata* leaf litter to increase soil nutrient availability in the years following successful invasion (*Rodgers et al., 2008*) also suggests that this species has evolved to compete effectively at high, rather than low, soil resources. In addition, we note that our experiment simulates competition between first year individuals of *A. petiolata* and competitor species. In the field, competition also takes place between spring flowering *A. petiolata* plants that have over wintered as rosettes and newly germinating plants of competitor species, which can further advantage *A. petiolata* (*Herold et al., 2011*). The observation that nutrient depletion caused by conspecifics reduces individual plant performance, however, corroborates previous findings that *A. petiolata* experiences relatively strong intra-specific competition (*Meekins & McCarthy, 1999*; *Davis et al., 2012*; *Leicht-Young, Pavlovic & Adams, 2012*), which would limit the net reproductive rate of established populations. The possibility of nutrient-limitation mediated density-dependent population regulation is consistent with recent demographic analyses showing that in situations where other biotic factors such as herbivory are excluded, established *A. petiolata* populations decline towards extinction (*Knight et al., 2009*; *Kalisz, Spigler & Horvitz, 2014*).

Growth form was the best predictor of the ability of competitor species to either resist or suppress *A. petiolata*, but this effect varied with soil history and competition metric. For example, *A. petiolata* suppressed the growth of all three growth forms in control soil, but this effect was more modest in *A. petiolata* history soil. By contrast, grasses suppressed

invader biomass in both soil treatments, whereas forbs had no effect, and trees appeared to facilitate the growth of *A. petiolata* in control soils. The ability of grasses to suppress *A. petiolata* may have occurred because they were taller than other growth forms at a young age, which would increase light acquisition (*Grime, 1977*; *Gaudet & Keddy, 1988*; *Goldberg & Landa, 1991*; *Rosch, Van Rooyen & Theron, 1997*; *Keddy et al., 2002*; *Wang et al., 2010*). Grasses also had relatively fine roots (Fig. 5), which would increase nutrient uptake capacity (*Aerts, Boot & Van der Aart, 1991*; *Goldberg, 1996*; *Casper & Jackson, 1997*). Nonetheless, height may be the most important factor because grasses and forbs had similar photosynthetic capacity and root architecture, yet forbs did not suppress *A. petiolata* biomass. *Meekins & McCarthy (1999)* also found that *A. petiolata* was a weaker competitor against tall relative to short species. Though the ability of *A. petiolata* to suppress tree growth has been previously observed (*Stinson et al., 2007*), the observation that trees might facilitate *A. petiolata* growth was unexpected. This effect may be due to aspects that were unique to the two tree species, *P. strobus* and *T. occidentalis*, which had the strongest beneficial effect on *A. petiolata*. These species were the only conifers in the sample and also ranked lowest in terms of growth rate (Fig. 1). The relatively strong growth form effects of competitor species on *A. petiolata* we report here may not be universal however. Other studies suggest that trees can be strong competitors (*Meekins & McCarthy, 1999*; *Smith & Reynolds, 2014*) and grasses can be weak competitors (*Smith & Reynolds, 2014*) against *A. petiolata*.

Functional trait variation, beyond that associated with growth form, did not predict either the ability of competitor species to resist suppression by, or their ability to suppress, *A. petiolata*. When growth form was included in multiple regressions between traits and competitive response or competitive effect, no significant relationships were found, regardless of soil treatment (Table 4). There was also limited trait plasticity in response to *A. petiolata* history in soil (Fig. 5), despite strong effects on plant biomass. These findings are consistent with those of *Wang et al. (2010)*, who also reported weak relationships between trait values and competitive ability. The inability to detect specific relationships between traits and competitive ability could be caused by the possibility that competitive ability depends on combinations of several traits or traits that were not measured (*Wardle et al., 1998*; *Wang et al., 2010*), or because functionally alternate strategies, such as efficient resource acquisition or resource storage, can result in similar competitive abilities (*Grime, 1977*).

Our findings have implications for recent hypotheses about how competitive response and competitive effect should be correlated across environments (*Keddy, Twolan-Strutt & Wisheu, 1994*; *Keddy et al., 2002*; *Wang et al., 2010*). Specifically, competitive response is expected to be context specific, varying with resource availability or other ecological and environmental factors, and is not expected to be correlated across environments. By contrast, competitive effect is expected to be a general property of a species, such that it is positively correlated across environments (*Wang et al., 2010*). Our results are generally consistent with these predictions (Fig. 6), but the relationship between competitive effect in control and history soils was weaker than (i.e., $r^2 = 0.12$, Fig. 6B) found in other studies (*Keddy, Twolan-Strutt & Wisheu, 1994*; *Keddy et al., 2002*; *Wang et al., 2010*). Observing such context dependency in the competitive effect of *A. petiolata* was not unique to our

study. For example, *Smith & Reynolds (2014)* found that *A. petiolata* could suppress other species under high light conditions, but had much weaker effects in the shade. Our findings suggest the ability of competitor species to either resist suppression by, or suppress, *A. petiolata* cannot be confidently predicted from one ecological context to another.

In conclusion, our findings suggest that *A. petiolata* has the potential to displace resident species in a community upon initial invasion via a relatively strong competitive ability. However, its competitive ability is weakened, rather than strengthened, by conspecific soil history effects. Like previous studies, we observed that soil with a history of *A. petiolata* reduces the ability of mycorrhizal fungi to colonize the roots of competitor species. However, this negative novel weapons effect on mycorrhizal plant species did not appear overcome the negative history effects of soil nutrient depletion on *A. petiolata*. These findings suggest that the inhibitory potential of *A. petiolata* on competitor species via mycorrhizal suppression may not be as strong as previously suggested. In addition, because longer term effects of *A. petiolata* invasion include an overall increase in plant available soil nitrogen and phosphorus (*Rodgers et al., 2008*), mycorrhizal suppression is unlikely to be a strong mechanism of competition in the years following invasion. This is because the effect of losing the fungal symbiont approaches neutrality when mycorrhizal plants are grown with supplemental nutrients (*Hoeksema et al., 2010*; *Johnson, 2010*). The potential for weak mycorrhizal suppression effects suggests that eradication or control measures based on minimizing novel biochemical weapons effects in *A. petiolata* may be less successful than other approaches. As suggested by other studies, reducing propagule pressure by removal of flowering individuals (*Herold et al., 2011*; *Phillips-Mao, Larson & Jordan, 2014*) and suppressing browsing by deer (*Kalisz, Spigler & Horvitz, 2014*) could be more effective strategies to counteract the invasion of *A. petiolata* in North America.

## ACKNOWLEDGEMENTS

We thank A Benoit, E Bothwell, CM Caruso, E Pacey, P Rekret, R Rivkin and three anonymous reviewers for comments on the manuscript. N Sokol and AM Benscoter assisted with various phases of the project.

### Funding

This research was supported by grants from the Ontario Early Researcher Award program, the Canada Foundation for Innovation and the Natural Sciences and Engineering Research Council of Canada. The funders had no role in study design, data collection and analysis, decision to publish, or preparation of the manuscript.

### Grant Disclosures

The following grant information was disclosed by the authors:
Ontario Early Researcher Award program.
Canada Foundation for Innovation.
Natural Sciences and Engineering Research Council of Canada.

## Competing Interests

The authors declare there are no competing interests.

## Author Contributions

- Gary T. Poon conceived and designed the experiments, performed the experiments, analyzed the data, wrote the paper, prepared figures and/or tables, reviewed drafts of the paper.
- Hafiz Maherali conceived and designed the experiments, analyzed the data, contributed reagents/materials/analysis tools, wrote the paper, prepared figures and/or tables, reviewed drafts of the paper.

## Supplemental Information

Supplemental information for this article can be found online at http://dx.doi.org/10.7717/peerj.1090#supplemental-information.

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
