# Peer review of "Competitive interactions between a nonmycorrhizal invasive plant, Alliaria petiolata, and a suite of mycorrhizal grassland, old field, and forest species"

_PeerJ, doi:10.7717/peerj.1090_

## Round 0.1 · original submission · Major Revisions

This was a very well written paper tackling an interesting topic of considerable conservation concern and ecological interest. The results potentially have important implications for both managers and ecological theory.

However, you will note that the views of the two reviewers were very divided. Reviewer 1 was concerned about the nature of your experimental design and, in particular, that the impact of Alliaria on soil nutrients is artificial and confounded with potential allelopathic effects. I shared these concerns to some extent but would like to give you the opportunity to address them. In your rebuttal letter please take care to address this issue in detail and explain how you have dealt with it in your revised submission. In addition to the reviewers' feedback I've made some further specific comments and corrections on the attached pdf.

Reviewer 1 ·

Basic reporting

No Comment

Experimental design

The major concern I have about this research is that it does not have a suitable control. The authors wanted to determine if Alliaria petiolata was more competitive against 27 mycorrhizal species of herbs (including native and introduced species), and mycorrhizal colonization when they were grown in soil with or without a legacy of A. petiolata. For the purpose of this study, the authors collected soil from a forested area that had no history of invasion by A. petiolata. To create the legacy effect, they planted seeds of the invasive species in one-half of the soil, which was contained in thirty 35L tubs. After five weeks, the seedlings were thinned to a density of 80 plants m-2 and grown for an additional 5 month and harvested.

Growing A. petiolata for a period of slightly longer than six months in the “legacy soil” and then removing the plant biomass from the soil resulted in a reduction of available inorganic nutrients (NO-3,NH4+, P, K, and Mg), because nutrients were removed from the soil and sequestered in the plant tissue. Consequently, soil inorganic nutrients in the “legacy soil” were lower than in the control soil without a legacy of A. petiolata. Of the organic nutrients tested, the greatest reduction occurred with the two nitrogen ions. Unfortunately, a reduction in nitrogen can have a strongly negative effect on the growth of garlic mustard and likely in the production of compounds that reduce mycorrhizal colonization, because these compounds have an energetic cost (Hewin and Hyatt, 2010, Biological Invasions 12: 2639-2647).

The authors tried to the balance the differences in soil nutrients by adding fertilizer once during the study, but nutrient differences between the two treatments remained. However, they do not report these differences in soil nutrients or when during the study additional nutrients were added. Other studies have dealt with these differences by periodic addition of inorganic nutrients (Callaway et al., 2008, Ecology 89:1041-1055) or by adding leaves of different plants, which potentially affect mycorrhizae differently, to the soil of the treatment and control (Allison et al. 2011, Ecosphere Volume 2(10), Article 110).

The authors did not indicate how the non-legacy soil was stored while the process of producing the “legacy soil” was occurring, and they should, because the conditions under which soil is stored could have an effect on the microbial community.

Validity of the findings

I agree with the authors that a case can be made that the reductions in A. petiolata competitiveness and biomass in legacy soil compared to “control soil” are due to the reduction of nitrogen in the legacy soil. However, a reduction in nitrogen was not a planned part of the study. Under field conditions the biomass of A. petiolata is not removed and a depletion of nitrogen does not occur as a result of the removal of biomass. However, the reduction in mycorrhizal colonization may be due to the longer period of time (5 weeks + 5 months) that the “legacy soil” had A. petiolata growing in it than the control soil. It would have been useful to obtain the Mycorrhizal Inoculum Potential of the legacy and control soil at the time seedlings were planted in both treatments.

Reviewer 2 ·

Basic reporting

This manuscript adheres to the structure and policies for PeerJ. It is well-written, clearly organized, and original primary research. This manuscript provides findings that are well-developed, scientifically sound, and should be of interest to the readers of PeerJ.

Experimental design

The research question being investigated & the experimental design here is relevant and rigorous. The following additional information within the methods would be helpful:
- Depth of soil collected from the field (deeper layers would not be valid to use in this study)
- Depth of the 35L tubs where A. petiolata was grown to create legacy soil & the rooting depth of the plants at the end of the soil training period (if deeper soils that were not contacted by the roots were mixed in with the top layers of soil, this should be clearly explained and justified)
- How many seeds were initially planted before thinning to 80 plants/m2?
- Why was 5 months of training chosen as the time frame - citation or reasoning? Is it long enough to accumulate allelopathic effects?
- Soil nutrient measurements were made, but values are not provided. These would be really helpful to include. Also, do you mean P, Mg and K ions or bulk density?
- Why was 63 days chosen as the growing period? Again, is this longer to see growth differences? Justification?

Validity of the findings

The experiments are controlled and statistically sound. Figures and tables are well-represented and clear. I make some suggestions for additional information to include below.

Additional comments

I think it would be helpful to explain a bit more (maybe with field observations as justification) the reasoning behind using no previous plant growth as the control for the legacy soils. Comparing soil grown with A. petioloata vs. soils grown with nothing can be viewed as experimentally problematic.

It would be helpful to clarify that within your experiments, you cannot distinguish between potential effects of allelopathy or reduced soil nutrients. Also, you did not measure the presence of allelopathic chemicals within your soils, so there is no way to know whether your soils created these conditions.

Within the discussion I think it would be helpful to address the issue of phenology with A. petiolata. In thinking about other mechanisms that may provide specific advantages, this appears to be an important one, along with the deer browsing. Within your experiment all plants were grown together at the same time, but this is not what happens out in the field.

I suggest possibly including the following 2 papers within the discussion:
Stinson et al. 2007 Northeastern Naturalist. Impacts of garlic mustard invasion on a forest understory community
Rodgers et al. 2008 Oecologica. The invasive species Alliaria petiolata (garlic mustard) increases soil nutrient availability in northern hardwood-conifer forests

- Line 123 - add "soil" before "resources"
- Move the description of pot size on lines 159-160 to line 152 where you first mention the pots
- Line 143 clarify that you mean A. petiolata "seedlings were transplanted"
- Line 224 "either resist" is written twice here - eliminate
- Line 320 eliminate "either" before "in legacy soil"
- Lines 324-325 - this first sentence of the discussion is unclear and vague. I suggest eliminating it
- Line 339 add "soils rather" before "than control soils"
- Line 383 eliminate "much" before " more"
- Lines 402-403 reword this sentence - awkward structure and a little unclear

---

## Round 0.2 · Minor Revisions

Many thanks for your considerable efforts to respond to the previous reviewers' comments and suggestions. Both reviewers seem to agree that the paper is now suitable for publication though Reviewer 2 has asked you to make a couple of clarifications.

In addition I would still like to see you be a little more circumspect in how you interpret your results and to be up-front about the implications of your experimental design. Whilst reading your revised version I made the following notes:

P4 L82: Change to "resistance to the novel biochemicals"

P4 L101: "Moreover, if A. petiolata suppresses mycorrhizal fungi while simultaneously depleting soil nutrients" - Here’s the issue for me. In your experiment the establishing garlic mustard plants have to deal with low nutrient conditions which appears to inhibit their growth. In the natural setting nutrients in above-ground plant biomass are returned to the soil at the end of each year which means nutrient reduction occurs over course of growing season. Competing plants do not, therefore, start the season in reduced nutrient conditions, this may develop over the course of the season if garlic mustard is a strong competitor. I think that means your experiment shows that competitive interactions vary over soil nutrient gradients but I’m not convinced your experiment mimics what happens in the real world. That’s not a barrier to publication but could you maybe address this issue somewhere?

P5 L104: "These effects should be ... dominant in the community" - Not clear to me why this process would operate more strongly once garlic mustard is dominant. In initial stages of invasion surely effect of garlic mustard on soil nutrients will be relatively minor? Is there not going to be some kind of tipping point at which invasion becomes complete enough to have significant effect on soil nutrients and at that point interaction with allelopathy will become very important as nutrients during growing season become somewhat limiting?

P5 L114: "during the establishment period" - As in seedling establishment period or period during which invasive becomes established in the community?

P6 L131: "The A. petiolata history soil treatment ... disadvantage competitor species" - Ok so this is the place where things perhaps need to be clarified. The “legacy” treatment is not what really happens during in invasion as the nutrients removed by garlic mustard growth were not returned to the soil. I think what you’ve done is create a low and high nutrient content (in relative terms) soils but used garlic mustard as your mechanism for removing nutrients. There may be legacy effects if allelopathic chemicals were created during the nutrient reduction treatment but you can’t determine if this was the case. I know it probably seems like I’m being pedantic but I think it would be useful because you’ll need to point out that a) your experiment does not strictly mimic what happens during invasion and b) that you study interactions across a soil nutrient gradient (in itself interesting). I realise you’ve dealt with this in the discussion but you need to be upfront about it.

P6 L137: "We predicted that A. petiolata ... than in control soil" - How about “We predicted that performance of mycorrhizal plant species would be inhibited in soil where nutrient content had previously been reduced by the growth of A. petiolate (hereafter referred to as “legacy soil”).” or something like that.

P15 L374: "Our results indicate that ... history of conspecific growth" - Make explicit here that effects may be due reduced nutrient conditions, allelopathy or both.

P16 L418: "The propensity for ... expected from resource competition theory" - How do you square this statement with the fact that nutrients are returned to the soil at end of the growing season and that previous studies have shown garlic mustard increases soil nutrients?

My apologies if these requests go over old-ground somewhat but I think the clarifications are necessary to adequately reflect the recommendations of the previous review.

Reviewer 2 ·

Basic reporting

No comments

Experimental design

The revised manuscript now justifies the limitations and interpretations of the study. Details for the methods that were previously missing are now available. The purpose of the study is better explained and the inclusion of resource competition theory coupled with loss of mycorrhizal symbionts is much more clear. Also, the allelopathic vs. nutrient differences are clarified.

Validity of the findings

The findings are now better explained and limitations to the study are well described.

Additional comments

Thank you for carefully responding to each concern of the reviewers. I believe the manuscript to be much stronger now.

Reviewer 3 ·

Basic reporting

This manuscript reports on the effects of Alliaria petiolata on both native and non-native tree, forb and grass species within an experimental system. The authors grew plants either with or without Alliaria and in soil with either a history of Alliaria invasion or without. The manuscript is well written and organized. The background information and information concerning methods and results is well written and easily followed. The paper constitutes an acceptable publication unit and is well grounded within the broader field of invasion biology especially as it relates to Alliaria invasion.

Experimental design

I thought this was a well designed experiment.

Validity of the findings

The findings are interesting and supported by the data. I did have questions about two aspects of the reported data.

First, the authors reported that nutrient content in forest soil after several months of growth with Alliaria (Alliaria history soil) contained significantly lower levels of many nutrients. But I wondered whether this comparison was made to the originally collected forest soil or to the forest soil that did not have Alliaria planted in it but was also incubated for several months? I would expect that nutrient content may have changed after several months of incubation in the absence of plants, but it was a little unclear if the nutrient comparison was made against the unplanted but incubated forest soil or the forest soil that was originally collected.

Second, Quercus macrocarpa was examined for the extent of arbuscular mycorrhizal (AM) colonization but as the authors indicate in table 1 this plant species should be expected to form relationships with ectomycorrhizal fungi. So it struck me as a strange choice for assessing AM colonization. Why not assess Prunus as the second tree species? It was interesting that you could detect AM fungi colonizing the root system in Q. macrocarpa, but I wonder whether the symbiotic relationship is out of context and what that means for our interpretation of how plants may perform under natural conditions. The colonization by AM mycorrhiza may be an artifact of the experimental system and also the use of newly germinated tree seedlings whose root systems could become colonized by a non-preferred fungal mutualist. Consequently, I think the results concerning this species should be interpreted cautiously and this limitation should be discussed in the paper.

Additional comments

The authors should check the table labeling. Right now table 1 (the plant species list) and table 2 (3-way ANOVA) are both labeled as table 1 in the table headings.

I thought this was a well done experiment and well written. I enjoyed reading the paper.

---

## Round 0.3 · accepted · Accept

Many thanks for taking the time to make the requested changes and clarifications and congratulations on completing an interesting piece of work.